# Constrained Policy Optimization via Bayesian World Models

**Yarden As** [*]
ETH Zurich

**Ilnura Usmanova, Sebastian Curi** [†]
ETH Zurich

**Andreas Krause**
ETH Zurich

## Abstract

Improving sample-efficiency and safety are crucial challenges when deploying reinforcement learning in high-stakes real world applications. We propose LAMBDA, a novel model-based approach for policy optimization in safety critical tasks modeled via constrained Markov decision processes. Our approach utilizes Bayesian world models, and harnesses the resulting uncertainty to maximize optimistic upper bounds on the task objective, as well as pessimistic upper bounds on the safety constraints. We demonstrate LAMBDA's state of the art performance on the Safety-Gym benchmark suite in terms of sample efficiency and constraint violation.

## 1 Introduction

A central challenge in deploying reinforcement learning (RL) agents in real-world systems is to avoid unsafe and harmful situations (Amodei et al., 2016). We thus seek agents that can learn efficiently and safely by acting cautiously and ensuring the safety of themselves and their surroundings.

A common paradigm in RL for modeling safety critical environments are constrained Markov decision processes (CMDP) (Altman, 1999). CMDPs augment the common notion of the reward by an additional cost function, e.g., indicating unsafe state-action pairs. By bounding the cost, one obtains bounds on the probability of harmful events. For the tabular case, CMDPs can be solved via Linear Programs (LP) (Altman, 1999). Most prior work for solving non-tabular CMDPs, utilize model-free algorithms (Chow et al., 2015; Achiam et al., 2017; Ray et al., 2019; Chow et al., 2019). Most notably, these methods are typically shown to asymptotically converge to a constraint-satisfying policy. However, similar to model-free approaches in unconstrained MDPs, these approaches typically have a very high sample complexity, i.e., require a large number of – potentially harmful – interactions with the environment.

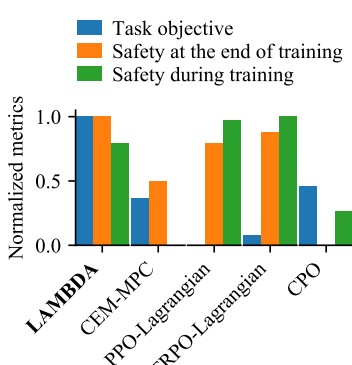

Figure 1: Normalized performance and safety metrics, averaged across tasks of the Safety-Gym (Ray et al., 2019) SG6 benchmark. LAMBDA achieves constraint satisfaction in all tasks of the SG6 benchmark while significantly improving performance and sample efficiency. See Appendix H for further details on normalization.

A promising alternative to improve sample efficiency is to use model-based reinforcement learning (MBRL) approaches (Deisenroth & Rasmussen, 2011; Chua et al., 2018; Hafner et al., 2019a; Janner et al., 2019). In particular, Bayesian approaches to MBRL quantify uncertainty in the estimated model (Depeweg et al., 2017), which can be used to guide exploration, e.g., via the celebrated *optimism in the face of uncertainty* principle (Brafman & Tennenholtz, 2003; Auer & Ortner, 2007; Curi et al., 2020a). While extensively explored in the unconstrained setting, Bayesian model-based deep RL approaches for solving general CMDPs remain largely unexplored. In this paper, we close this gap by proposing LAMBDA, a Bayesian approach to model-based policy optimization in CMDPs. LAMBDA learns a safe policy by experiencing unsafe as well as rewarding events through *model-generated* trajectories instead of real ones. Our main contributions are as follows:

---

[*]Correspondence to: `yardas@ethz.ch`
[†]Equal contribution.

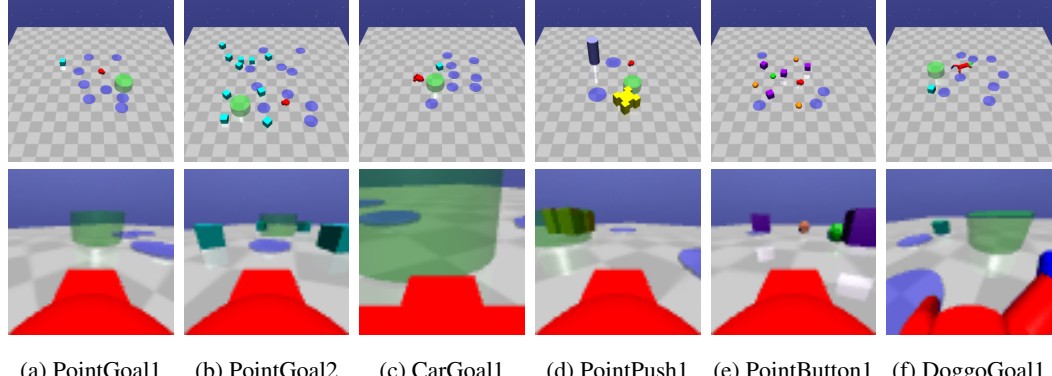

|     (a) PointGoal1 | (b) PointGoal2 | (c) CarGoal1 | (d) PointPush1 | (e) PointButton1 | (f) DoggoGoal1 |

Figure 2: Safety-Gym SG6 benchmark tasks. In our experiments, we use first-person-view images of size 64×64 pixels as observations. Green objects represent goals that should be reached by the robot. Apart from the yellow box, that should be pushed to the goal area in the PointPush1 task, hitting all other types of objects is considered an unsafe behavior. In Safety-Gym, stochasticity is achieved by performing a random number of simulation steps before exposing the agent with a new observation.

- We show that it is possible to perform constrained optimization by back-propagating gradients of both task and safety objectives through the world-model. We use the Augmented Lagrangian (Nocedal & Wright, 2006; Li et al., 2021) approach for constrained optimization.
- We harness the probabilistic world model to trade-off between optimism for exploration and pessimism for safety.
- We empirically show that LAMBDA successfully solves the SG6 benchmark tasks in Safety-Gym (Ray et al., 2019) benchmark suite with first-person-view observations as illustrated in Figure 2. Furthermore, we show that LAMBDA outperforms other model-based and model-free methods in this benchmark.

## 2    RELATED WORK

**Interpretations of safety in RL research**    We first acknowledge that there exist different definitions for safety (García et al., 2015). One definition uses reversible Markov decision processes (Moldovan & Abbeel, 2012) that, informally, define safety as reachability between states and use backup policies (Eysenbach et al., 2017) to ensure safety. Another definition adopts robust Markov decision processes (Nilim & Ghaoui, 2005; Tamar et al., 2013; Tessler et al., 2019), which try to maximize performance under the worst-case transition model. Finally, in this work we use non-tabular CMDPs (Altman, 1999) that define the safety requirements as a cost function that should be bounded by a predefined threshold. Similarly, Achiam et al. (2017); Dalal et al. (2018); Ray et al. (2019); Chow et al. (2019); Stooke et al. (2020); Bharadhwaj et al. (2021); Turchetta et al. (2021) use non-tabular CMDPs as well, but utilize model-free techniques together with function approximators to solve the constrained policy optimization problem. As we further demonstrate in our experiments, it is possible to achieve better sample efficiency with model-based methods.

**Safe model-based RL**    A successful approach in applying Bayesian modeling to low-dimensional continuous-control problems is to use Gaussian Processes (GP) for model learning. Notably, Berkenkamp et al. (2017) use GPs to construct confidence intervals around Lyapunov functions which are used to optimize a policy such that it is always within a Lyapunov-stable region of attraction. Furthermore, Koller et al. (2018); Wabersich & Zeilinger (2021) use GP models to certify the safety of actions within a model predictive control (MPC) scheme. Likewise, Liu et al. (2021) apply MPC for high-dimensional continuous-control problems together with ensembles of neural networks (NN), the Cross Entropy Method (CEM) (Kroese et al., 2006) and rejection sampling to maximize expected returns of safe action sequences. However, by planning online only for short horizons and not using critics, this method can lead to myopic behaviors, as we later show in our experiments.

Lastly, similarly to this work, Zanger et al. (2021) use NNs and constrained model-based policy optimization. However, they do not use model uncertainty within an optimistic-pessimistic framework but rather to limit the influence of erroneous model predictions on their policy optimization. Even though using accurate model predictions can accelerate policy learning, this approach does not take advantage of the epistemic uncertainty (e.g., through optimism and pessimism) when such accurate predictions are rare.

**Exploration, optimism and pessimism** Curi et al. (2021) and Derman et al. (2019) also take a Bayesian optimistic-pessimistic perspective to find robust policies. However, these approaches do not use CMDPs and generally do not explicitly address safety. Similarly, Bharadhwaj et al. (2021) apply pessimism for conservative safety critics, and use them for constrained *model-free* policy optimization. Finally, Efroni et al. (2020) lay a theoretical foundation for the exploration-exploitation dilemma and optimism in the setting of tabular CMPDs.

## 3 PRELIMINARIES

### 3.1 SAFE MODEL-BASED REINFORCEMENT LEARNING

**Markov decision processes** We consider an episodic Markov decision process with discrete time steps $t \in \{0, \ldots, T\}$. We define the environment's state as $s_t \in \mathbb{R}^n$ and an action taken by the agent, as $a_t \in \mathbb{R}^m$. Each episode starts by sampling from the initial-state distribution $s_0 \sim \rho(s_0)$. After observing $s_0$ and at each step thereafter, the agent takes an action by sampling from a policy distribution $a_t \sim \pi(\cdot|s_t)$. The next state is then sampled from an unknown transition distribution $s_{t+1} \sim p(\cdot|s_t, a_t)$. Given a state, the agent observes a reward, generated by $r_t \sim p(\cdot|s_t, a_t)$. We define the performance of a pair of policy $\pi$ and dynamics $p$ as

$$J(\pi, p) = \mathbb{E}_{a_t \sim \pi, s_{t+1} \sim p, s_0 \sim \rho} \left[ \sum_{t=0}^{T} r_t \big| s_0 \right]. \tag{1}$$

**Constrained Markov decision processes** To make the agent adhere to human-defined safety constraints, we adopt the constrained Markov decision process formalism of Altman (1999). In CMPDs, alongside with the reward, the agent observes cost signals $c_t^i$ generated by $c_t^i \sim p(\cdot|s_t, a_t)$ where $i = 1, \ldots, C$ denote the distinct unsafe behaviors we want the agent to avoid. Given $c_t^i$, we define the constraints as

$$J^i(\pi, p) = \mathbb{E}_{a_t \sim \pi, s_{t+1} \sim p, s_0 \sim \rho} \left[ \sum_{t=0}^{T} c_t^i \big| s_0 \right] \leq d^i, \ \forall i \in \{1, \ldots, C\}, \tag{2}$$

where $d^i$ are human-defined thresholds. For example, a common cost function is $c^i(s_t) = \mathbf{1}_{s_t \in \mathcal{H}^i}$, where $\mathcal{H}^i$ is the set of harmful states (e.g., all the states in which a robot hits an obstacle). In this case, the constraint (2) can be interpreted as a bound on the probability of visiting the harmful states. Given the constraints, we aim to find a policy that solves, for the true unknown dynamics $p^\star$,

$$\max_{\pi \in \Pi} J(\pi, p^\star) \quad \text{s.t.} \quad J^i(\pi, p^\star) \leq d^i, \ \forall i \in \{1, \ldots, C\}. \tag{3}$$

**Model-based reinforcement learning** Model-based reinforcement learning revolves around the repetition of an iterative process with three fundamental steps. First, the agent gathers observed transitions (either by following an initial policy, or an offline dataset) of $\{s_{t+1}, s_t, a_t\}$ into a dataset $\mathcal{D}$. Following that, the dataset is used to fit a statistical model $p(s_{t+1}|s_t, a_t, \theta)$ that approximates the true transition distribution $p^\star$. Finally, the agent uses the statistical model for planning, either within an online MPC scheme or via offline policy optimization. In this work, we consider the cost and reward functions as unknown and similarly to the transition distribution, we fit statistical models for them as well. Modeling the transition density allows us to cheaply generate synthetic sequences of experience through the factorization $p(s_{\tau:\tau+H}|s_{\tau-1}, a_{\tau-1:\tau+H-1}, \theta) = \prod_{t=\tau}^{\tau+H} p(s_{t+1}|s_t, a_t, \theta)$ whereby $H$ is a predefined sequence horizon (see also Appendix I). Therefore, as already shown empirically (Deisenroth & Rasmussen, 2011; Chua et al., 2018; Hafner et al., 2019b), MBRL achieves superior sample efficiency compared to its model-free counterparts. We emphasize that in safety-critical tasks, where human supervision during training might be required, sample efficiency is particularly important since it can reduce the need for human supervision.

**Learning a world model**  Aiming to tighten the gap between RL and real-world problems, we relax the typical full observability assumption and consider problems where the agent receives an observation $o_t \sim p(\cdot|s_t)$ instead of $s_t$ at each time step. To infer the transition density from observations, we base our world model on the Recurrent State Space Model (RSSM) introduced in Hafner et al. (2019a). To solve this inference task, the RSSM approximates the posterior distribution $s_{\tau:\tau+H} \sim q_\phi(\cdot|o_{\tau:\tau+H}, a_{\tau-1:\tau+H-1})$ via an inference network parameterized by $\phi$. We utilize the inference network to filter the latent state as new observations arrive, and use the inferred latent state as the policy's input. Furthermore, the RSSM models the predictive distribution $p(s_{\tau:\tau+H}|s_{\tau-1}, a_{\tau-1:\tau+H-1}, \theta)$ as a differentiable function. This property allows us to perform the constrained policy optimization by backpropagating gradients through the model. We highlight that the only requirement for the world model is that it models $p(s_{\tau:\tau+H}|s_{\tau-1}, a_{\tau-1:\tau+H-1}, \theta)$ as a differentiable function. Hence, it is possible to use other architectures for the world model, as long as $p(s_{\tau:\tau+H}|s_{\tau-1}, a_{\tau-1:\tau+H-1}, \theta)$ is differentiable (e.g., see GP-models in Curi et al. (2020b)).

## 3.2 A Bayesian view on model-based reinforcement learning

**The Bayesian predictive distribution**  A common challenge in MBRL is that policy optimization can "overfit" by exploiting inaccuracies of the estimated model. A natural remedy is to quantify uncertainty in the model, to identify in which situations the model is more likely to be wrong. This is especially true for safety-critical tasks in which model inaccuracies can mislead the agent into taking unsafe actions. A natural approach to uncertainty quantification is to take a Bayesian perspective, where we adopt a prior on the model parameters, and perform (approximate) Bayesian inference to obtain the posterior. Given such a posterior distribution over the model parameters $p(\theta|\mathcal{D})$, we marginalize over $\theta$ to get a Bayesian predictive distribution. Therefore, by maintaining such posterior, $p(s_{\tau:\tau+H}|s_{\tau-1}, a_{\tau-1:\tau+H-1})$ is determined by the law of total probability

$$p(s_{\tau:\tau+H}|s_{\tau-1}, a_{\tau-1:\tau+H-1}) = \mathbb{E}_{\theta \sim p(\theta|\mathcal{D})}\left[p(s_{\tau:\tau+H}|s_{\tau-1}, a_{\tau-1:\tau+H-1}, \theta)\right]. \qquad (4)$$

Such Bayesian reasoning forms the basis of celebrated MBRL algorithms such as PETS (Chua et al., 2018) and PILCO (Deisenroth & Rasmussen, 2011), among others.

**Maintaining a posterior over model parameters**  Since exact Bayesian inference is typically infeasible, we rely on approximations. In contrast to the popular approach of bootstrapped ensembles (Lakshminarayanan et al., 2017) widely used in RL research, we use the Stochastic Weight Averaging-Gaussian (SWAG) approximation (Maddox et al., 2019) of $p(\theta|\mathcal{D})$. Briefly, SWAG constructs a posterior distribution over model parameters by performing moment-matching over iterates of stochastic gradient decent (SGD) (Robbins, 2007). The main motivation behind this choice is the lower computational and memory footprint of SWAG compared to bootstrapped ensembles. This consideration is crucial when working with parameter-rich world models (such as the RSSM). Note that our approach is flexible and admits other variants of approximate Bayesian inference, such as variational techniques (Graves, 2011; Kingma & Welling, 2014).

# 4 Lagrangian Model-based Agent (LAMBDA)

We first present our proposed approach for planning, i.e., given a world model representing a CMDP, how to use it to solve the CMPD. Following that, we present our proposed method for finding the optimistic and pessimistic models with which the algorithm solves the CMDP before collecting more data.

## 4.1 Solving CMPDs with model-based constrained policy optimization

**The Augmented Lagrangian**  We first consider the optimization problem in Equation (3). To find a policy that solves Equation (3), we take advantage of the Augmented Lagrangian with proximal relaxation method as described by Nocedal & Wright (2006). First, we observe that

$$\max_{\pi \in \Pi} \min_{\lambda \geq 0} \left[ J(\pi) - \sum_{i=1}^{C} \lambda^i \left( J^i(\pi) - d^i \right) \right] = \max_{\pi \in \Pi} \begin{cases} J(\pi) & \text{if } \pi \text{ is feasible} \\ -\infty & \text{otherwise} \end{cases} \qquad (5)$$

is an equivalent form of Equation (3). Hereby $\lambda^i$ are the Lagrange multipliers, each corresponding to a safety constraint measured by $J^i(\pi)$. In particular, if $\pi$ is feasible, we have $J^i(\pi) \leq d^i \ \forall i \in$

$\{1, \ldots, C\}$ and so the maximum over $\lambda^i$ is satisfied if $\lambda^i = 0 \ \forall i \in \{1, \ldots, C\}$. Conversely, if $\pi$ is infeasible, at least one $\lambda^i$ can be arbitrarily large to solve the problem in Equation (5). Particularly, Equation (5) is non-smooth when $\pi$ transitions between the feasibility and infeasibility sets. Thus, in practice, we use the following relaxation:

$$\max_{\pi \in \Pi} \min_{\boldsymbol{\lambda} \geq 0} \left[ J(\pi) - \sum_{i=1}^{C} \lambda^i \left( J^i(\pi) - d^i \right) + \frac{1}{\mu_k} \sum_{i=1}^{C} \left( \lambda^i - \lambda_k^i \right)^2 \right], \quad (*)$$

where $\mu_k$ is a non-decreasing penalty term corresponding to gradient step $k$. Note how the last term in $(*)$ encourages $\lambda^i$ to stay *proximal* to the previous estimate $\lambda_k^i$ and as a consequence making $(*)$ a smooth approximation of the left hand side term in Equation (5). Differentiating $(*)$ with respect to $\lambda^i$ and substituting back leads to the following update rule for the Lagrange multipliers:

$$\forall i \in \{1, \ldots, C\} : \lambda_{k+1}^i = \begin{cases} \lambda_k^i + \mu_k (J^i(\pi) - d^i) & \text{if } \lambda_k^i + \mu_k (J^i(\pi) - d^i) \geq 0 \\ 0 & \text{otherwise} \end{cases} . \quad (6)$$

We take gradient steps of the following unconstrained objective:

$$\tilde{J}(\pi; \boldsymbol{\lambda}_k, \mu_k) = J(\pi) - \sum_{i=1}^{C} \Psi^i(\pi; \lambda_k^i, \mu_k), \quad (7)$$

where

$$\Psi^i(\pi; \lambda_k^i, \mu_k) = \begin{cases} \lambda_k^i (J^i(\pi) - d^i) + \frac{\mu_k}{2} \left( J^i(\pi) - d^i \right)^2 & \text{if } \lambda_k^i + \mu_k (J^i(\pi) - d^i) \geq 0 \\ -\frac{(\lambda_k^i)^2}{2\mu_k} & \text{otherwise.} \end{cases} \quad (8)$$

**Task and safety critics** For the task and safety critics, we use the reward value function $v(\boldsymbol{s}_t)$ and the cost value function of each constraint $v^i(\boldsymbol{s}_t) \ \forall i \in \{1, \ldots, C\}$. We model $v(\boldsymbol{s}_t)$ and $v^i(\boldsymbol{s}_t)$ as neural networks with parameters $\psi$ and $\psi^i$ respectively. Note that we omit here the dependency of the critics in $\pi$ to reduce the notational clutter. As in Hafner et al. (2019b), we use TD($\lambda$) (Sutton & Barto, 2018) to trade-off the bias and variance of the critics with bootstrapping and Monte-Carlo value estimation. To learn the task and safety critics, we minimize the following loss function

$$\mathcal{L}_{v_\psi^\pi}(\psi) = \mathbb{E}_{\pi, p(\boldsymbol{s}_{\tau:\tau+H} | \boldsymbol{s}_{\tau-1}, \boldsymbol{a}_{\tau-1:\tau+H-1}, \theta)} \left[ \frac{1}{2H} \sum_{t=\tau}^{\tau+H} \left( v_\psi^\pi(\boldsymbol{s}_t) - V_\lambda(\boldsymbol{s}_t) \right)^2 \right] \quad (9)$$

which uses the model to generate samples from $p(\boldsymbol{s}_{\tau:\tau+H} | \boldsymbol{s}_{\tau-1}, \boldsymbol{a}_{\tau-1:\tau+H-1}, \theta)$. We denote $v_\psi^\pi$ as the neural network that approximates its corresponding value function and $V_\lambda$ as the TD($\lambda$) value as presented in Hafner et al. (2019b). Similarly, $V_\lambda^i$ is the TD($\lambda$) value of the $i^{th}$ constraint $\forall i \in \{1, \ldots, C\}$. Note that, while we show here only the loss for the task critic, the loss function for the safety critics is equivalent to Equation (9).

**Policy optimization** We model the policy as a Gaussian distribution via a neural network with parameters $\xi$ such that $\pi_\xi(\boldsymbol{a}_t | \boldsymbol{s}_t) = \mathcal{N}(\boldsymbol{a}_t; NN_\xi^\mu(\boldsymbol{s}_t), NN_\xi^\sigma(\boldsymbol{s}_t))$. We sample a sequence $\boldsymbol{s}_{\tau:\tau+H} \sim p(\boldsymbol{s}_{\tau:\tau+H} | \boldsymbol{s}_{\tau-1}, \boldsymbol{a}_{\tau-1:\tau+H-1}, \theta)$, utilizing $\pi_\xi$ to generate actions on the fly (see Appendix I). To estimate $\tilde{J}(\pi; \boldsymbol{\lambda}_k, \mu_k)$, we compute $V_\lambda$ and $V_\lambda^i$ for each state in the sampled sequence. This approximation leads to the following loss function for policy learning

$$\mathcal{L}_{\pi_\xi}(\xi) = \mathbb{E}_{\pi_\xi, p(\boldsymbol{s}_{\tau:\tau+H} | \boldsymbol{s}_{\tau-1}, \boldsymbol{a}_{\tau-1:\tau+H-1}, \theta)} \left[ \frac{1}{H} \sum_{t=\tau}^{\tau+H} -V_\lambda(\boldsymbol{s}_t) \right] + \sum_{i=1}^{C} \Psi^i(\pi_\xi; \lambda_k^i, \mu_k). \quad (10)$$

We approximate $J^i(\pi_\xi)$ in $\Psi^i(\pi_\xi; \lambda_k^i, \mu_k)$ (recall Equation (8)) as

$$J^i(\pi_\xi) \approx \mathbb{E}_{\pi_\xi, p(\boldsymbol{s}_{\tau:\tau+H} | \boldsymbol{s}_{\tau-1}, \boldsymbol{a}_{\tau-1:\tau+H-1}, \theta)} \left[ \frac{1}{H} \sum_{t=\tau}^{\tau+H} V_\lambda^i(\boldsymbol{s}_t) \right] . \quad (11)$$

Moreover, the first expression in the policy's loss (10) approximates $J(\pi_\xi)$. We backpropagate through $p(\boldsymbol{s}_{\tau:\tau+H} | \boldsymbol{s}_{\tau-1}, \boldsymbol{a}_{\tau-1:\tau+H-1}, \theta)$ using path-wise gradient estimators (Mohamed et al., 2020).

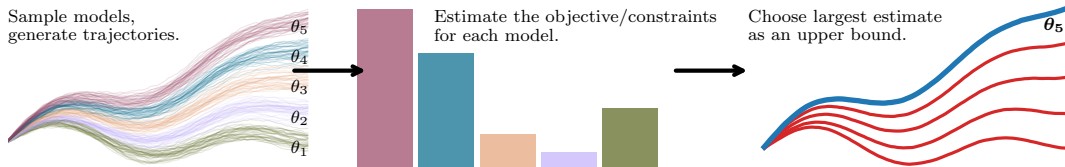

Figure 3: **Posterior sampling**: We sample $j = 1, \ldots, N$ models $\theta_j \sim p(\theta|\mathcal{D})$ (e.g., in this illustration, $N = 5$). For each model, we simulate trajectories that are conditioned on the same policy and initial state. **Objective and constraints**: For a given posterior sample $\theta_j$, we use the simulated trajectories to estimate $J(\pi, p_{\theta_j})$ and $J^i(\pi, p_{\theta_j})$ $\forall i \in \{1, \ldots, C\}$ with their corresponding critics. **Upper bounds**: Choose largest estimate for each of $J(\pi, p_{\theta_j})$ and $J^i(\pi, p_{\theta_j})$ $\forall i \in \{1, \ldots, C\}$ among their $N$ realizations.

## 4.2 ADOPTING OPTIMISM AND PESSIMISM TO EXPLORE IN CMDPs

**Optimistic and pessimistic models** As already noted by Curi et al. (2020a), greedily maximizing the expected returns by averaging over posterior samples in Equation (4) is not necessarily the best strategy for exploration. In particular, this greedy maximization does not deliberately leverage the *epistemic* uncertainty to guide exploration. Therefore, driven by the concepts of *optimism in the face of uncertainty* and upper confidence reinforcement learning (UCRL) (Auer et al., 2009; Curi et al., 2020a), we define a set of statistically plausible transition distributions denoted by $\mathcal{P}$ and let $p_\theta \in \mathcal{P}$ be a particular transition density in this set. Crucially, we assume that the true model $p^\star$ is within the support of $\mathcal{P}$, and that by sampling $\theta \sim p(\theta|\mathcal{D})$ the conditional predictive density satisfies $p(\boldsymbol{s}_{t+1}|\boldsymbol{s}_t, \boldsymbol{a}_t, \theta) = p_\theta \in \mathcal{P}$. These assumptions lead us to the following constrained problem

$$\max_{\pi \in \Pi} \max_{p_\theta \in \mathcal{P}} J(\pi, p_\theta) \quad \text{s.t.} \quad \max_{p_{\theta^i} \in \mathcal{P}} J^i(\pi, p_{\theta^i}) \leq d^i, \ \forall i \in \{1, \ldots, C\}. \tag{12}$$

The main intuition here is that jointly maximizing $J(\pi, p_\theta)$ with respect to $\pi$ and $p_\theta$ can lead the agent to try behaviors with potentially high reward due to *optimism*. On the other hand, the *pessimism* that arises through the inner maximization term is crucial to enforce the safety constraints. Being only optimistic can easily lead to dangerous behaviors, while being pessimistic can lead the agent to not explore enough. Consequently, we conjecture that optimizing for an optimistic task objective $J(\pi, p_\theta)$ combined with pessimistic safety constraints $J^i(\pi, p_{\theta^i})$ allows the agent to explore better task-solving behaviors while being robust to model uncertainties.

**Estimating the upper bounds** We propose an algorithm that estimates the objective and constraints in Equation (12) through sampling, as illustrated in Figure 3. We demonstrate our method in Algorithm 1. Importantly, we present Algorithm 1 only with the notation of the task critic. However, we use it to approximate the model's upper bound for each critic independently, i.e., for the costs as well as the reward value function critics.

---

**Algorithm 1** Upper confidence bounds estimation via posterior sampling

---

**Require:** $N, p(\theta|\mathcal{D}), p(\boldsymbol{s}_{\tau:\tau+H}|\boldsymbol{s}_{\tau-1}, \boldsymbol{a}_{\tau-1:\tau+H-1}, \theta), \boldsymbol{s}_{\tau-1}, \pi(\boldsymbol{a}_t, |\boldsymbol{s}_t)$.
 1: Initialize $\mathcal{V} = \{\}$           # Set of objective estimates, under different posterior samples.
 2: **for** $j = 1$ to $N$ **do**
 3:    $\theta \sim p(\theta|\mathcal{D})$.           # Posterior sampling (e.g., via SWAG).
 4:    $\boldsymbol{s}_{\tau:\tau+H} \sim p(\boldsymbol{s}_{\tau:\tau+H}|\boldsymbol{s}_{\tau-1}, \boldsymbol{a}_{\tau-1:\tau+H-1}, \theta)$ .      # Sequence sampling, see Appendix I.
 5:    Append $\mathcal{V} \leftarrow \mathcal{V} \cup \sum_{t=\tau}^{\tau+H} \mathrm{V}_\lambda(\boldsymbol{s}_t)$.
 6: **end for**
 7: **return** $\max \mathcal{V}$.

---

**The LAMBDA algorithm** Algorithm 2 describes how all the previously shown components interact with each other to form a model-based policy optimization algorithm. For each update step, we sample a batch of $B$ sequences with length $L$ from a replay buffer to train the model. Then, we sample $N$ models from the posterior and use them to generate novel sequences with horizon $H$ from every state in the replay buffer sampled sequences.

---

**Algorithm 2** LAMBDA

---

1: Initialize $\mathcal{D}$ by following a random policy or from an offline dataset.
2: **while** not converged **do**
3:     **for** $u = 1$ to $U$ update steps **do**
4:         Sample $B$ sequences $\{(\boldsymbol{a}_{\tau'-1:\tau'+L-1}, \boldsymbol{o}_{\tau':\tau'+L}, r_{\tau':\tau'+L}, c^i_{\tau':\tau'+L})\} \sim \mathcal{D}$ uniformly.
5:         Update model parameters $\theta$ and $\phi$.     # E.g., see Hafner et al. (2019a) for the RSSM.
6:         Infer $\boldsymbol{s}_{\tau':\tau'+L} \sim q_\phi(\cdot|\boldsymbol{o}_{\tau:\tau+L}, \boldsymbol{a}_{\tau'-1:\tau'+L-1})$.
7:         Compute $\sum_{t=\tau}^{\tau+H} \mathrm{V}_\lambda(\boldsymbol{s}_t), \sum_{t=\tau}^{\tau+H} \mathrm{V}^i_\lambda(\boldsymbol{s}_t)$ via Algorithm 1. Use each state in $\boldsymbol{s}_{\tau':\tau'+L}$ as an initial state for sequence generation.
8:         Update $\psi$ and $\psi^i$ via Equation (9) with $\sum_{t=\tau}^{\tau+H} \mathrm{V}_\lambda(\boldsymbol{s}_t)$ and $\sum_{t=\tau}^{\tau+H} \mathrm{V}^i_\lambda(\boldsymbol{s}_t)$.
9:         Update $\xi$ according to Equation (10) with $\sum_{t=\tau}^{\tau+H} \mathrm{V}_\lambda(\boldsymbol{s}_t)$ and $\sum_{t=\tau}^{\tau+H} \mathrm{V}^i_\lambda(\boldsymbol{s}_t)$.
10:       Update $\lambda^i$ via Equations (6) and (11).
11:     **end for**
12:     **for** $t = 1$ to $T$ **do**
13:         Infer $\boldsymbol{s}_t \sim q_\phi(\cdot|\boldsymbol{o}_t, \boldsymbol{a}_{t-1}, \boldsymbol{s}_{t-1})$.
14:         Sample $\boldsymbol{a}_t \sim \pi_\xi(\cdot|\boldsymbol{s}_t)$.
15:         Take action $\boldsymbol{a}_t$, observe $r_t, c^i_t, \boldsymbol{o}_{t+1}$ received from the environment.
16:     **end for**
17:     Update dataset $\mathcal{D} \leftarrow \mathcal{D} \cup \{\boldsymbol{o}_{1:T}, \boldsymbol{a}_{1:T}, r_{1:T}, c^i_{1:T}\}$.
18: **end while**

---

## 5   Experiments

We conduct our experiments with the SG6 benchmark as described by Ray et al. (2019), aiming to answer the following questions:

- How does our model-based approach compare to model-free variants in terms of performance, sample efficiency and constraint violation?

- What is the effect of replacing our proposed policy optimization method with an online planning method? More specifically, how does LAMBDA's policy compare to CEM-MPC of Liu et al. (2021)?

- How does our proposed optimism-pessimism formulation compare to greedy exploitation in terms of performance, sample efficiency and constraint violation?

We provide an open-source code for our experiments, including videos of the trained agents at https://github.com/yardenas/la-mbda.

### 5.1   SG6 Benchmark

**Experimental setup**   In all of our experiments with LAMBDA, the agent observes $64{\times}64$ pixels RGB images, taken from the robot's point-of-view, as shown in Figure 2. Also, since there is only one safety constraint, we let $J_c(\pi) \equiv J^1(\pi)$. We measure performance with the following metrics, as proposed in Ray et al. (2019):

- Average undiscounted episodic return for $E$ episodes: $\hat{J}(\pi) = \frac{1}{E} \sum_{i=1}^{E} \sum_{t=0}^{T_{\mathrm{ep}}} r_t$

- Average undiscounted episodic cost return for $E$ episodes: $\hat{J}_c(\pi) = \frac{1}{E} \sum_{i=1}^{E} \sum_{t=0}^{T_{\mathrm{ep}}} c_t$

- Normalized sum of costs during training, namely the *cost regret*: for a given number of total interaction steps $T$, we define $\rho_c(\pi) = \frac{\sum_{t=0}^{T} c_t}{T}$ as the cost regret.

We compute $\hat{J}(\pi)$ and $\hat{J}_c(\pi)$ by averaging the sum of costs and rewards across $E = 10$ evaluation episodes of length $T_{\mathrm{ep}} = 1000$, without updating the agent's networks and discarding the interactions made during evaluation. In contrast to the other metrics, to compute $\rho_c(\pi)$, we sum the costs accumulated *during training* and not evaluation episodes. The results for all methods are recorded once the agent reached 1M environment steps for PointGoal1, PointGoal2, CarGoal1 and 2M steps

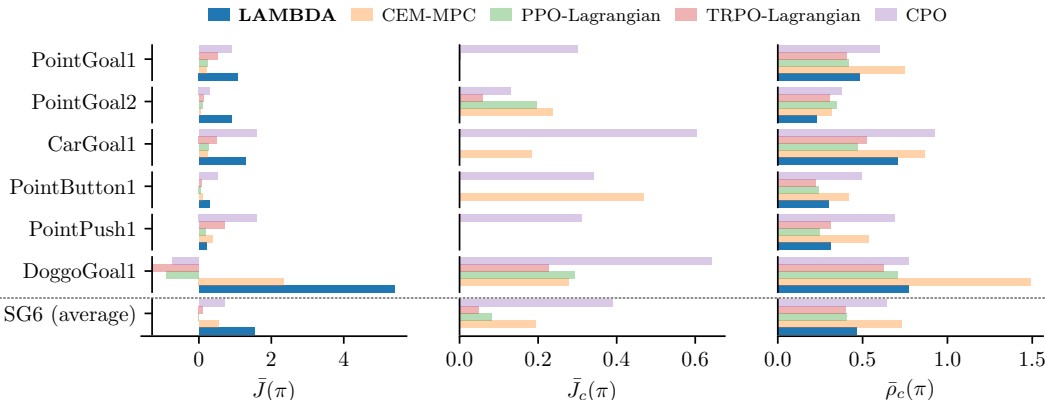

Figure 4: Experimental results for the SG6 benchmark. As done in Ray et al. (2019), we normalize the metrics and denote $\bar{J}(\pi), \bar{J}_c(\pi), \bar{\rho}_c(\pi)$ as the normalized metrics (see also Appendix H). We note that LAMBDA achieves better performance while satisfying the constraints during test time. Furthermore, LAMBDA attains similar result to the baseline in terms of the cost regret metric. We also note that CPO performs similarly to LAMBDA in solving some of the tasks but fails to satisfy the constraints in all tasks.

for PointButton1, PointPush2, DoggoGoal1 environments respectively. To reproduce the scores of TRPO-Lagrangian, PPO-Lagrangian and CPO, we follow the experimental protocol of Ray et al. (2019). We give further details on our experiment with CEM-MPC at the ablation study section.

**Practical aspects on runtime**    The simulator's integration step times are 0.002 and 0.004 seconds for the Point and Car robots, making each learning task run roughly 30 and 60 minutes in real life respectively. The simulator's integration step time of the Doggo robot is 0.01 seconds thereby training this task in real life would take about 167 minutes. These results show that in principle, the data acquisition time in real life can be very short. However in practice, the main bottleneck is the gradient step computation which takes roughly 0.5 seconds on a single unit of Nvidia GeForceRTX2080Ti GPU. In total, it takes about 18 hours to train an agent for 1M interaction steps, assuming we take 100 gradient steps per episode. In addition, for the hyperparameters in Appendix C, we get a total of 12M simulated interactions used for policy and value function learning per episode.

**Results**    The results of our experiments are summarized in Figure 4. As shown in Figure 4, LAMBDA is the only agent that satisfies the safety constraints in all of the SG6 tasks. Furthermore, thanks to its model-based policy optimization method, LAMBDA requires only 2M steps to successfully solve the DoggoGoal1 task and significantly outperform the other approaches. In Appendix B we examine further LAMBDA's sample efficiency compared to the model-free baseline algorithms. Additionally, in the PointGoal2 task, which is denser and harder to solve in terms of safety, LAMBDA significantly improves over the baseline algorithms in all metrics. Moreover, in Figure 1 we compare LAMBDA's ability to trade-off between the average performance and safety metrics across all of the SG6 tasks. One main shortcoming of LAMBDA is visible in the Point-Push1 environment where the algorithm fails to learn to push to box to the goal area. We attribute this failure to the more strict partial observability of this task and further analyse it in Appendix D.

## 5.2 Ablation study

**Unsafe LAMBDA**    As our first ablation, we remove the second term in Equation (10) such that the policy's loss only comprises the task objective (i.e., with only optimistic exploration). We make the following observations: (**1**) Both LAMBDA and unsafe LAMBDA solve the majority of the SG6 tasks, depending on their level of partial observability; (**2**) in the PointButton1, PointGoal1 and CarGoal1, LAMBDA achieves the same performance of unsafe LAMBDA, while satisfying the safety constraints; (**3**) LAMBDA is able to reach similar performance to unsafe LAMBDA in the PointGoal2 task which is strictly harder than the other tasks in terms of safety, as shown in Figure 5. We note that partial observability is present in all of the SG6 tasks due to the restricted field of view

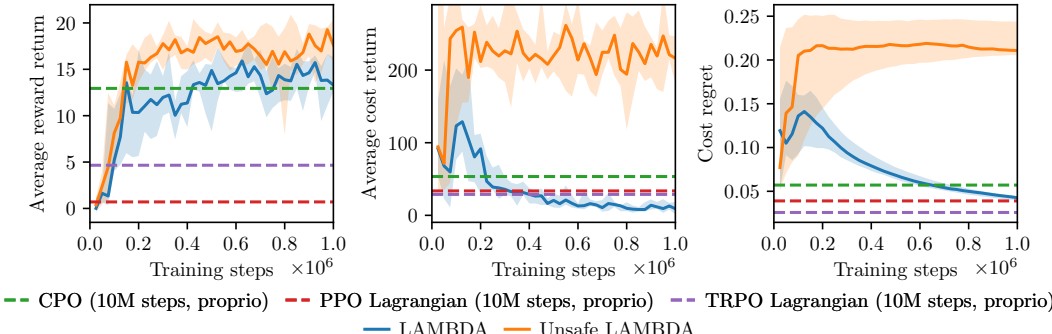

Figure 5: Learning curves of LAMBDA and its unsafe version on the PointGoal2 environment. As shown, LAMBDA exhibits similar performance in solving the task while maintaining the safety constraint and significantly improving over the baseline algorithms.

of the robot; i.e., the goal and obstacles are not always visible to the agent. In the DoggoGoal1 task, where the agent does not observe any information about the joint angles, LAMBDA is still capable of learning complex walking locomotion. However, in the PointPush1 task, LAMBDA struggles to find a task-solving policy, due to the harder partial observability of this task (see Appendix D). We show the learning curves for this experiment in Appendix E.

**Ablating policy optimization**    Next, we replace our actor-critic procedure and instead of performing policy optimization, we implement the proposed MPC method of Liu et al. (2021). Specifically, for their policy, Liu et al. (2021) suggest to use the CEM to maximize rewards over action sequences while rejecting the unsafe ones. By utilizing the same world model, we are able to directly compare the planning performance for both methods. As shown in Figure 11, LAMBDA performs considerably better than CEM-MPC. We attribute the significant performance difference to the fact that the CEM-MPC approach does not use any critics for its policy, making its policy short-sighted and limits its ability to address the credit assignment problem.

**Optimism and pessimism compared to greedy exploitation**    Finally, we compare our upper bounds estimation procedure with the more common approach of greedily maximizing the expected performance. More specifically, instead of employing Algorithm 1, we use Monte-Carlo sampling to estimate Equation (4) by generating trajectories with $N = 5$ sampled models and taking the average trajectory over the samples. By doing so, we get an estimate of the mean trajectory over sampled models together with the intrinsic stochasticity of the environment. We report our experiment results in Appendix G and note that LAMBDA is able to find safer and more performant policies than its greedy version.

## 6    CONCLUSIONS

We introduce LAMBDA, a Bayesian model-based policy optimization algorithm that conforms to human-specified safety constraints. LAMBDA uses its Bayesian world model to generate trajectories and estimate an optimistic bound for the task objective and pessimistic bounds for the constraints. For policy search, LAMBDA uses the Augmented Lagrangian method to solve the constrained optimization problem, based on the optimistic and pessimistic bounds. In our experiments we show that LAMBDA outperforms the baseline algorithms in the Safety-Gym benchmark suite in terms of sample efficiency as well as safety and task-solving metrics. LAMBDA learns its policy directly from observations in an end-to-end fashion and without prior knowledge. However, we believe that introducing prior knowledge with respect to the safety specifications (e.g., the mapping between a state and its cost) can improve LAMBDA's performance. By integrating this prior knowledge, LAMBDA can potentially learn a policy that satisfies constraints only from its model-generated experience and without ever violating the constraints in the real world. This leads to a notable open question on how to integrate this prior knowledge within the world model such that the safety constraints are satisfied *during* learning and not only at the end of the training process.

## 7    ACKNOWLEDGMENTS AND DISCLOSURE OF FUNDING

This project has received funding from the European Research Council (ERC) under the European Union's Horizon 2020 research and innovation programme grant agreement No 815943, the Swiss National Science Foundation under NCCR Automation, grant agreement 51NF40 180545 and the Swiss National Science Foundation, under grant SNSF 200021 172781.

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

## A   LEARNING CURVES FOR THE SG6 BENCHMARK

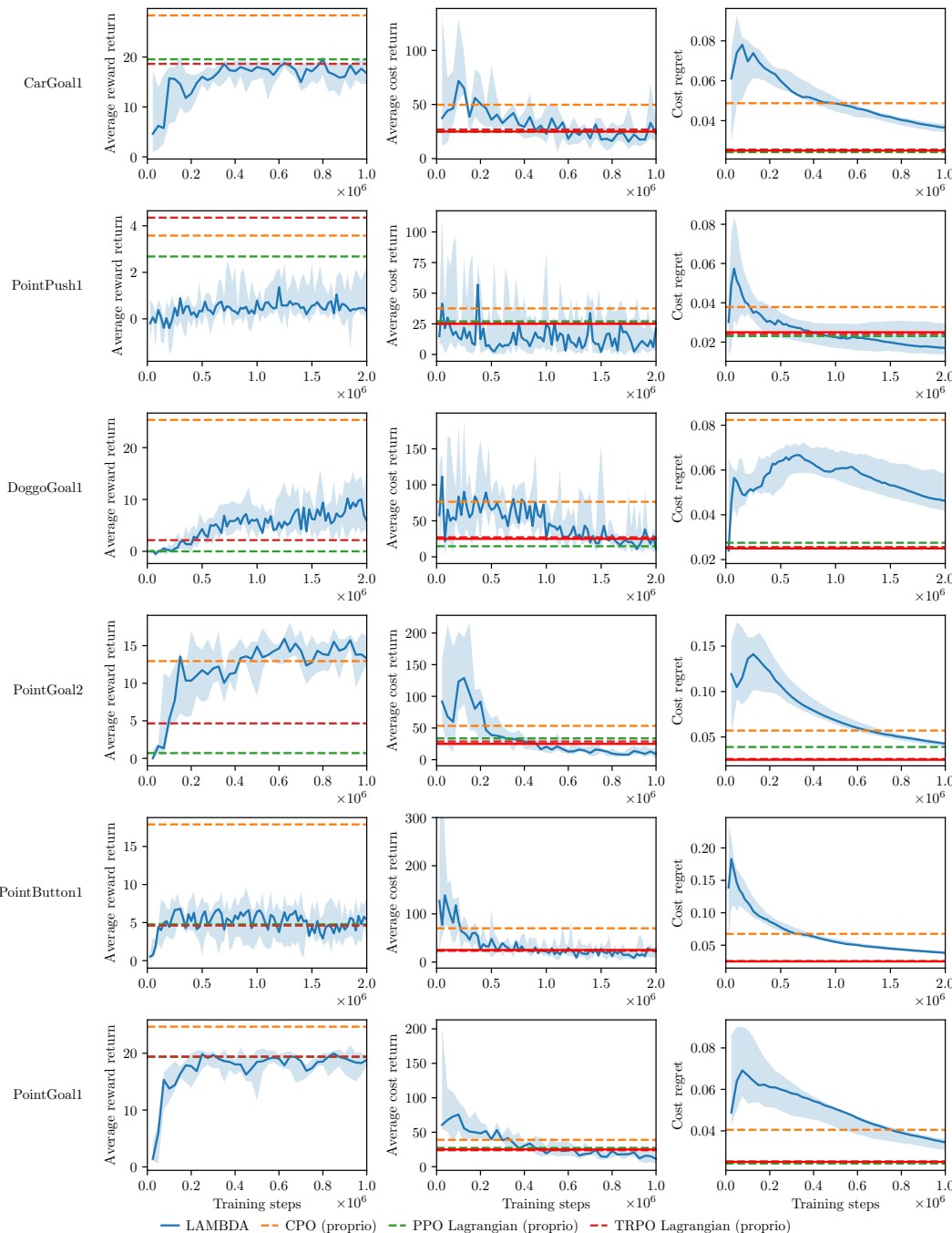

Figure 6: Benchmark results of LAMBDA. Solid red lines indicate the threshold value $d = 25$. Dashed lines correspond to the benchmark results for the baseline algorithms after 10M training steps for all tasks except the DoggoGoal1, which is trained for 100M environment steps, as in Ray et al. (2019). Shaded areas represent the 5% and 95% confidence intervals across 5 different random seeds.

# B  SAMPLE EFFICIENCY

We record LAMBDA's performance $\hat{J}(\pi)$ at the end of training and find the average number of steps (across seeds) required by the baseline model-free methods to match this performance. We demonstrate LAMBDA's sample efficiency with the ratio of the amount of steps required by the baseline methods and amount of steps at the end of LAMBDA's training. As shown in Figure 7, in the majority of tasks, LAMBDA is substantially more sample efficient. In the PointPush1 task LAMBDA is outperformed by the baseline algorithms, however we assign this to the partial observability of this task, as further analyzed in Appendix D. It is important to note that by taking LAMBDA's performance at the end of training, we take a conservative approach, as convergence can occur with many less steps, as shown in Figure 6.

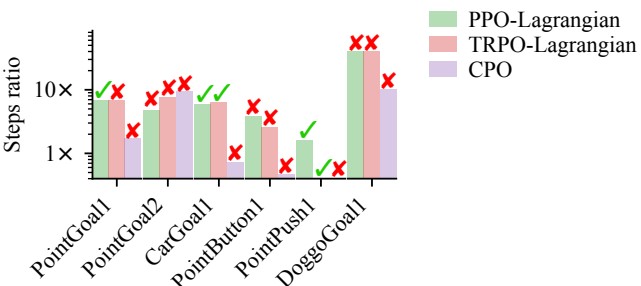

Figure 7: LAMBDA solves most of the tasks with significantly less interactions with the environment, compared to the baseline model-free methods. Green check marks and red 'x' indicate constraint satisfaction after reaching the required number of steps.

## C  HYPERPARAMETERS

In this section we specify the most prominent hyperparameters for LAMBDA, however we encourage the reader to visit `https://github.com/yardenas/la-mbda` as it holds many specific (but important) implementation details. Table 1 summarizes the hyperparameters used in our algorithm.

**World model**  As mentioned before, for our world model we use the RSSM proposed in Hafner et al. (2019a). Therefore, most of the architectural design choices (e.g., specific details of the convolutional layers) are based on the work there.

**Cost function**  We exploit our prior knowledge of the cost function's structure in the Safety-Gym tasks. Since we know a-priori that the cost function is implemented as an indicator function, our neural network approximation for it is modeled as a binary classifier. As training progresses, the dataset becomes less balanced as the agent observes less unsafe states. To deal with this issue, we give higher weight to the unsafe interactions in the binray classification loss.

**Posterior over model parameters**  in supervised learning, SWAG is typically used under the assumption of a fixed dataset $\mathcal{D}$, such that weight averaging takes place only during the last few training epochs. To adapt SWAG to the dataset's distributional shifts that occur during training in RL, we use an exponentially decaying running average. This allows us to maintain a posterior even within early stages of learning, as opposed to the original design of SWAG that normally uses the last few iterates of SGD to construct the posterior. Furthermore, we use a cyclic learning rate (Izmailov et al., 2019; Maddox et al., 2019) to help SWAG span over different regions of the weight space.

**Policy**  The policy outputs actions such that each element in $\boldsymbol{a}_t$ is within the range $[-1, 1]$. However, since we model $\pi_\xi(\boldsymbol{a}_t|\boldsymbol{s}_t)$ as $\mathcal{N}(\boldsymbol{a}_t; \mathrm{NN}_\xi^\mu(\boldsymbol{s}_t), \mathrm{NN}_\xi^\sigma(\boldsymbol{s}_t))$, we perform squashing of the Gaussian distribution into the range $[-1, 1]$, as proposed in Haarnoja et al. (2019) and Hafner et al. (2019b) by transforming it through a tangent hyperbolic bijector. We scale the actions in this way so that the model, which takes the actions as an input is more easily optimized. Furthermore, to improve numerical stability, the standard deviation term $\mathrm{NN}_\xi^\sigma(\boldsymbol{s}_t)$ is passed through a softplus function.

**Value functions**  To ensure learning stability for the critics, we maintain a shadow instance for each value function which is used in the bootstrapping in Equation (9). We clone the shadow instance such that it lags $U$ update steps behind its corresponding value function. Furthermore, we stop gradient on $\mathrm{V}_\lambda$ when computing the loss function in Equation (9).

**General**  Parameter update for all of the neural networks is done with ADAM optimizer (Kingma & Ba, 2014). We use ELU (Clevert et al., 2016) as activation function for all of the networks except for the convolutional layers in the world model in which we use ReLU (Nair & Hinton, 2010).

Table 1: Hyperparameters for LAMBDA. For other safety tasks, we recommend first tuning the initial Lagrangian, penalty and penalty power factor at different scales and then fine-tune the safety discount factor to improve constraint satisfaction. We emphasize that it is possible to improve the performance of each task separately by fine-tuning the hyperparameters on a per-task basis.

| Name | Symbol | Value | Additional |
|---|---|---|---|
| World model | | | |
| Batch size | $B$ | 32 | |
| Sequence length | $L$ | 50 | |
| Learning rate | | 1e-4 | |
| SWAG | | | |
| Burn-in steps | | 500 | Steps before weight averaging starts. |
| Period steps | | 200 | Use weights every 'Period steps' to update weights running average. |
| Models | | 20 | Averaging buffer length. |
| Decay | | 0.8 | Exponential averaging decay factor. |
| Cyclic LR factor | | 5.0 | End cyclic lr period with the base LR times this factor. |
| Posterior samples | $N$ | 5 | |
| Safety | | | |
| Safety critic learning rate | | 2e-4 | |
| Initial penalty | $\mu_0$ | 5e-9 | |
| Initial Lagrangian | $\lambda_0^1$ | 1e-6 | |
| Penalty power factor | | 1e-5 | Multiply $\mu_k$ by this factor at each gradient step. |
| Safety discount factor | | 0.995 | |
| General | | | |
| Update steps | $U$ | 100 | |
| Critic learning rate | | 8e-5 | |
| Policy learning rate | | 8e-5 | |
| Action repeat | | 2 | Repeat same action for this amount of steps. |
| Discount factor | | 0.99 | |
| TD($\lambda$) factor | $\lambda$ | 0.95 | |
| Sequence generation horizon | $H$ | 15 | |

## D    POINTPUSH1 ENVIRONMENT WITH A TRANSPARENT BOX

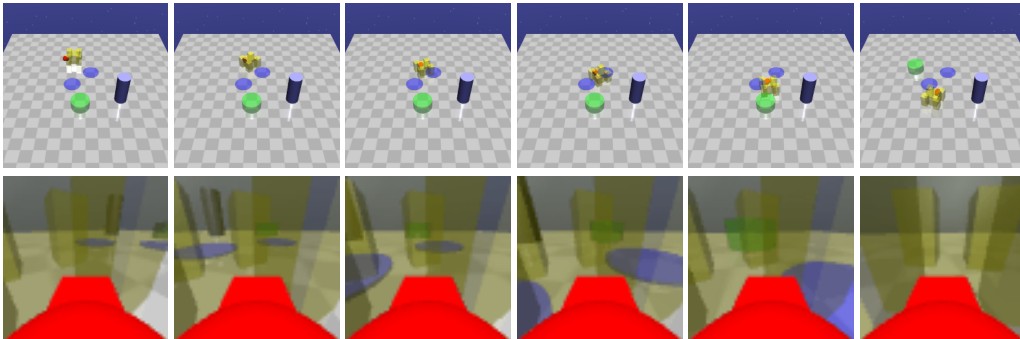

Figure 8: PointPush1 with partially transparent box. When the color of the box is solid LAMBDA struggles in solving the task due to occlusion of the goal by the box. By changing the transparency of the box, we make the PointPush1 task less partially observable and thus easier to solve.

In all of our experiments with the PointPush1 task, we provide the agent image observations in which the box is observed as a solid non-transparent object. As previously shown, we note that in this setting, LAMBDA *and* unsafe LAMBDA fail to learn the task. We maintain that this failure arises from the fact that this task is significantly harder in terms of partial observability, as the goal is occluded by the box while the robot pushes the box.[1] Furthermore, in Ray et al. (2019), the authors eliminate issues of partial observability by using what they term as "pseudo-LiDAR" which is not susceptible to occlusions as it can see through objects. To verify our hypothesis, we change the transparency of the box such that the goal is visible through it, as shown in Figure 8 and compare LAMBDA's performance with the previously tested PointPush1 task. We present the learning curves of the experiments in Figure 9. As shown, LAMBDA is able to safely solve the PointPush1 task if

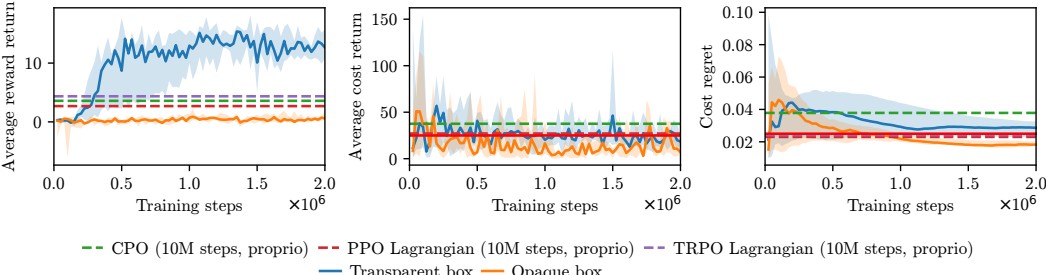

Figure 9: Learning curves for the PointPush1 task and ObservablePointPush1 task which uses partially transparent box. We also show the baseline algorithms performance with a "pseudo-LiDAR" observation.

the goal is visible through the box. We conclude that the PointPush1 task makes an interesting test case for future research on partially observable environments.

---

[1]Please see https://github.com/yardenas/la-mbda for a video illustration.

# E  UNSAFE LAMBDA

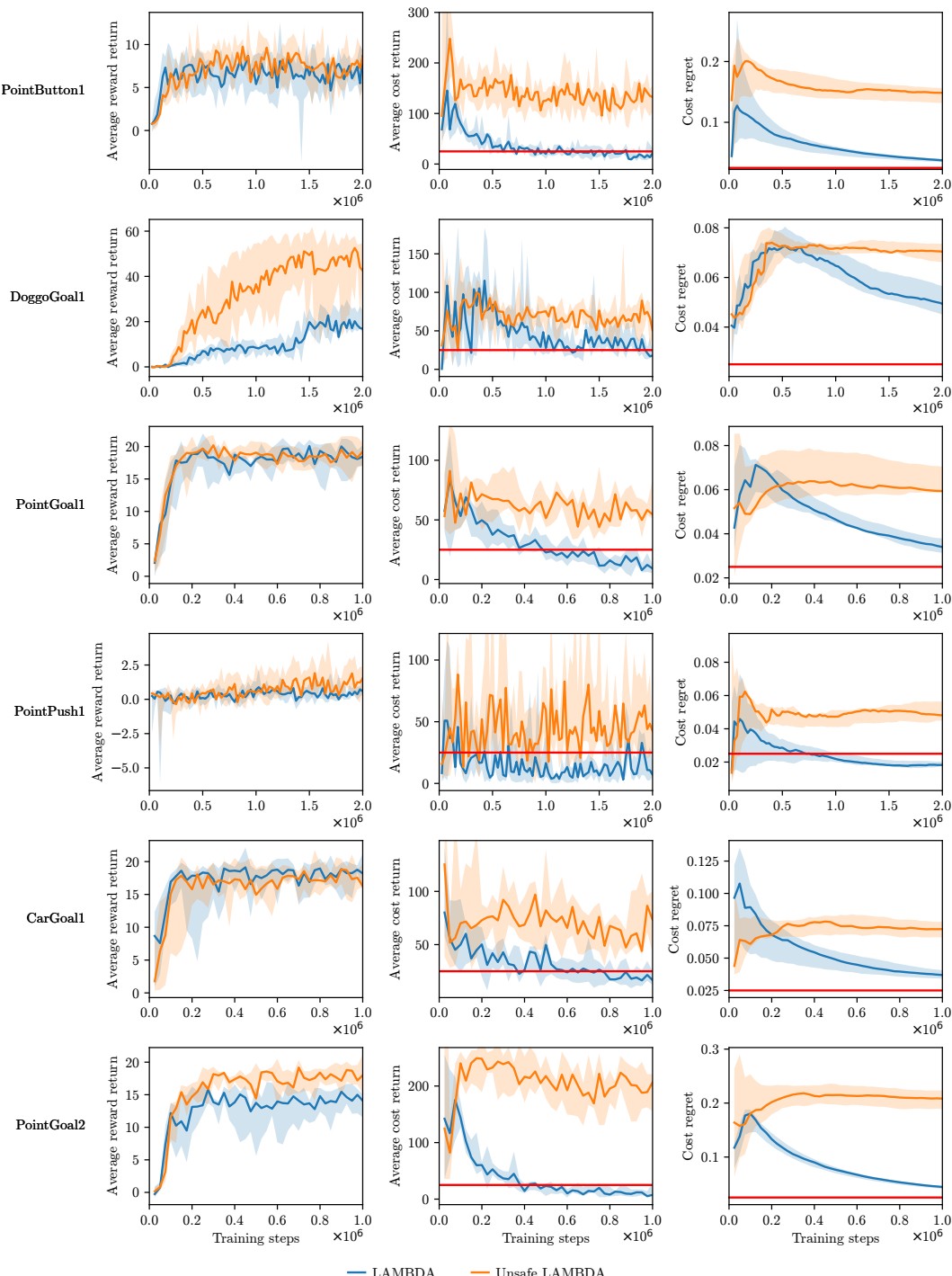

Figure 10: Benchmark results of LAMBDA and its unsafe implementation. In the majority of the tasks, LAMBDA is able to find policies that perform similarly to the unsafe version while satisfying the constraints. Interestingly, apart from the DoggoGoal1 and PointGoal2 tasks LAMBDA's policies are able to achieve similar returns while learning to satisfy the constraints.

# F COMPARISON WITH CEM-MPC

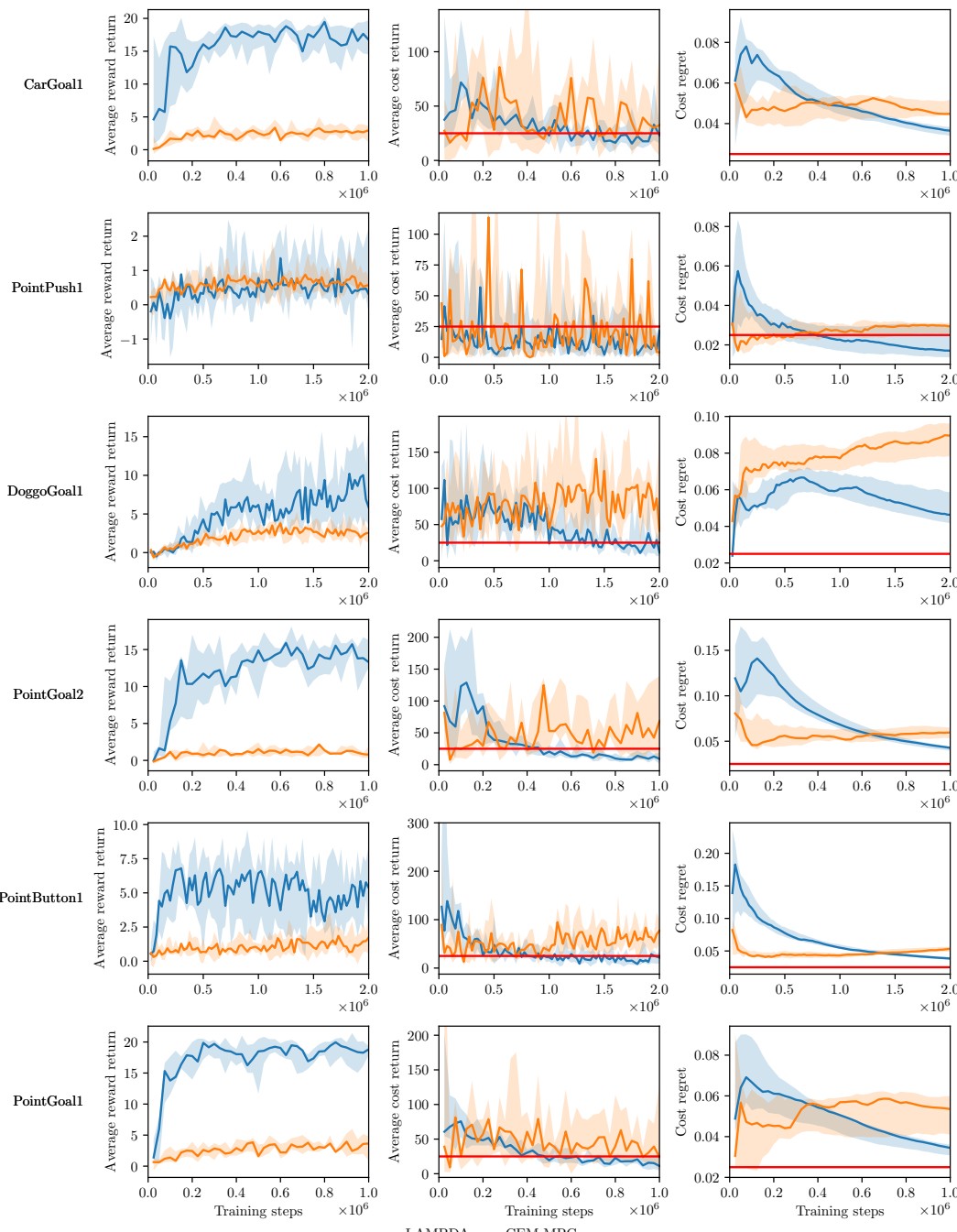

Figure 11: Comparison of LAMBDA when ablating the policy optimization and using CEM-MPC with rejection sampling as introduced in Liu et al. (2021). As shown, LAMBDA performs substantially better than CEM-MPC. We believe that when the goal is not visible to the agent, CEM-MPC's policy fails to locate it and drive the robot to it. On the contrary, in our experiments, we observed that LAMBDA typically rotates until the goal becomes visible, thus allowing the robot to gather significantly more goals.

## G  OPTIMISM AND PESSIMISM COMPARED TO GREEDY EXPLOITATION

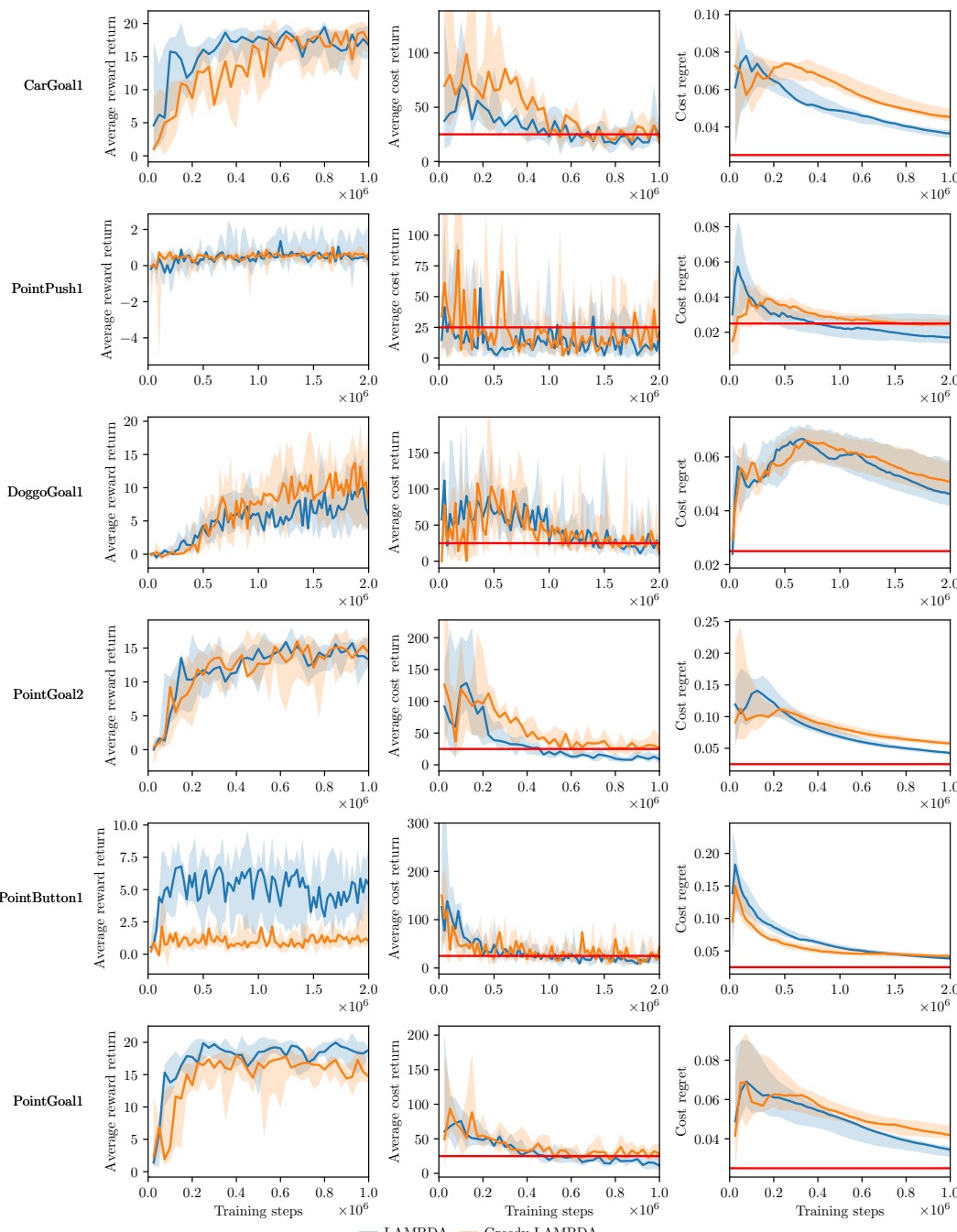

Figure 12: Comparison of LAMBDA and greedy exploitation. LAMBDA generally solves that tasks with better performance and with improved sample efficiency. Notably, the greedy exploitation variant fails to solve the the PointButton1 task.

## H    COMPARING ALGORITHMS IN THE SG6 BENCHMARK

To get a summary of how different algorithms behave across all of the SG6 tasks, we follow the proposed protocol in Ray et al. (2019). That is, we obtain "characteristic metrics" for each of the environments by taking the metrics recorded at the end of training of an unconstrained PPO agent (Schulman et al., 2017). We denote these metrics as $\hat{J}^{\text{PPO}}, \hat{J}_c^{\text{PPO}}$ and $\hat{\rho}_c^{\text{PPO}}$. We then normalize the recorded metrics for each environment as follows:

$$\bar{J}(\pi) = \frac{\hat{J}(\pi)}{\hat{J}^{\text{PPO}}}$$

$$\bar{J}_c(\pi) = \frac{\max(0, \hat{J}_c(\pi) - d)}{\max(10^{-6}, \hat{J}_c^{\text{PPO}} - d)} \tag{13}$$

$$\bar{\rho}_c(\pi) = \frac{\rho_c(\pi)}{\rho_c^{\text{PPO}}}.$$

By performing this normalization with respect to the performance of PPO, we can take an average of each metric across all of the SG6 environments.

To produce Figure 1, we scale the normalized metrics to $[0, 1]$, such that for each metric, the best performing algorithm attains a score of 1.0 and the worst performing algorithm attains a score of 0.

## I  BACKPROPAGATING GRADIENTS THROUGH A SEQUENCE

---

**Algorithm 3** Sampling from the predictive density $p_\theta(\boldsymbol{s}_{\tau:\tau+H}|\boldsymbol{s}_{\tau-1}, \boldsymbol{a}_{\tau-1:\tau+H-1}, \theta)$

---

**Require:** $\pi_\xi(\boldsymbol{a}_t|\boldsymbol{s}_t), p_\theta(\boldsymbol{s}_{t+1}|\boldsymbol{s}_t, \boldsymbol{a}_t), \boldsymbol{s}_{\tau-1}$
1: **for** $t = \tau - 1$ to $\tau + H$ **do**
2:   $\boldsymbol{a}_t \sim \pi_\xi(\cdot|\text{stop\_gradient}(\boldsymbol{s}_t))$    # Stop gradient from $\boldsymbol{s}_t$ when conditioning the policy on it.
3:   $\boldsymbol{s}_{t+1} \sim p(\cdot|\boldsymbol{s}_t, \boldsymbol{a}_t, \theta)$
4: **end for**
5: **return** $\boldsymbol{s}_{\tau:\tau+H}$

---

We use the reparametrization trick (Kingma & Welling, 2014) to compute gradients through sampling procedures as both $\pi_\xi$ and $p_\theta$ are modeled as normal distributions. Backpropagating gradients through the model can be easily implemented with modern automatic differentiation tools.

Importantly, we stop the gradient computation in Algorithm 3 when conditioning the policy on $\boldsymbol{s}_{t-1}$. We do so to avoid any recurrent connections between an action $\boldsymbol{a}_{t-1}$ and the preceding states to $\boldsymbol{s}_t$ such that eventually, backpropagation to actions occurs only from their dependant succeeding values. We further illustrate this in Figure 13.

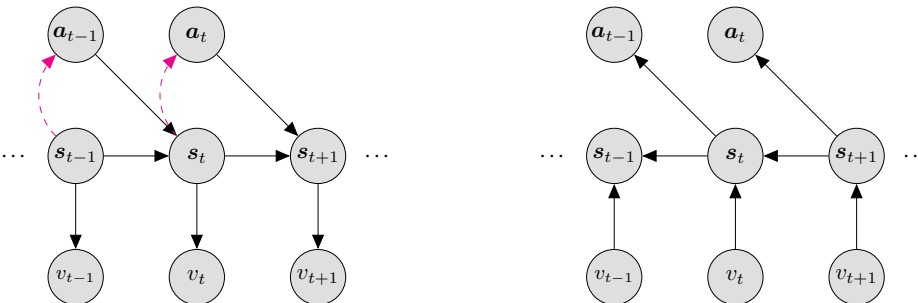

(a) Forward pass of trajectory sampling. We stop gradient flow on the magenta dashed arrows.

(b) Backward pass from values to their inducing actions.

Figure 13: Computational graphs for the backward and forward passes of Algorithm 3.

# J    SCORES FOR SG6

The .json format files, summarizing the scores for the experiments of this work are available at
https://github.com/yardenas/la-mbda.

Table 2: LAMBDA's unnormalized scores for the SG6 tasks.

|  | $\hat{J}(\pi)$ | $\hat{J}_c(\pi)$ | $\rho_c(\pi)$ |
|---|---|---|---|
| PointGoal1 | 18.822 | 11.200 | 0.034 |
| PointGoal2 | 13.300 | 9.100 | 0.043 |
| CarGoal1 | 16.745 | 23.100 | 0.036 |
| PointPush1 | 0.314 | 21.400 | 0.017 |
| PointButton1 | 5.372 | 21.700 | 0.038 |
| DoggoGoal1 | 5.867 | 11.400 | 0.046 |
| Average | 10.07 | 16.317 | 0.0360 |

Table 3: Experiment results for the SG6 benchmark. We present the results with the tuple $(\bar{J}(\pi), \bar{J}_c(\pi), \bar{\rho}_c(\pi))$ of the normalized metrics.

|  | TRPO-Lagrangian | PPO-Lagrangian | CPO | LAMBDA |
|---|---|---|---|---|
| PointGoal1 | 0.51, 0.004, **0.405** | 0.24, **0.0**, 0.419 | 0.898, 0.302, 0.599 | **1.077**, **0.0**, 0.483 |
| PointGoal2 | 0.119, 0.059, 0.304 | 0.09, 0.197, 0.349 | 0.306, 0.132, 0.377 | **0.902**, **0.0**, **0.229** |
| CarGoal1 | 0.501, **0.0**, 0.522 | 0.255, **0.0**, 0.474 | **1.579**, 0.604, 0.924 | 1.284, **0.0**, 0.704 |
| PointPush1 | 0.714, **0.0**, 0.315 | 0.185, **0.0**, **0.249** | **1.606**, 0.311, 0.687 | 0.203, **0.0**, 0.309 |
| PointButton1 | 0.077, **0.0**, **0.223** | 0.058, **0.0**, 0.242 | **0.516**, 0.343, 0.495 | 0.287, **0.0**, 0.302 |
| DoggoGoal1 | -1.257, 0.227, **0.624** | -0.891, 0.293, 0.707 | -0.723, 0.643, 0.769 | **5.400**, **0.0**, 0.770 |
| SG6 (average) | 0.111, 0.048, **0.399** | -0.011, 0.082, 0.407 | 0.697, 0.389, 0.642 | **1.526**, **0.0**, 0.466 |

