# OpenReview forum: "Constrained Policy Optimization via Bayesian World Models"
_ICLR.cc/2022/Conference — ICLR 2022 Spotlight_

### Official Review · Reviewer_HHZf · 2021-10-25

**Correctness:** 4
**Technical Novelty And Significance:** 3
**Empirical Novelty And Significance:** 4
**Recommendation:** 8
**Confidence:** 4

**Details Of Ethics Concerns:**

I do not have any ethical concerns.

**Main Review:**

First, I will summarize the Pros and Cons of the paper entitled with "Constrained Policy Optimization via Bayesian World Models."

### Pros
- New and promising paradigm. Utilizing Bayesian world models for CMDPs is indeed powerful tool for RL community. Since humans would decide their behavior using such way of thinking in real life; thus, the idea for using Bayesian world models for CMDPs would be worthwhile to study.
- Experiments. The effectiveness of the proposed method is clearly demonstrated in SG6 tasks in Safety-Gym compared with reasonable baselines.
- Well-written texts. It is easy to understand the key ideas in this paper. Also, related work is fully surveyed and introduced.
- Source-code is nice. If it is open-sourced, it would be quite helpful for the research community. (I just read through it. I did not run the code on my environment.)

### Cons
- Given there are many papers on Bayesian world models for "unconstrained" MDP, the ideas in this paper are little-bit straight forward. Incorporating optimism for reward and pessimism for safety is not a new idea, which is not an essential contribution.

### Questions
Although this paper is clearly presented and I don't have many questions, there are several unclear points.
- Q1: The world model (i.e., RSSM) is trained from scratch in the authors' experiments? My understanding is that, in the experiments presented in the paper, the world model is trained from scratch. If a pre-trained model is obtained, the training of the agent will become much safer. Is my understanding correct?
- Q2: Though it may be trivial, necessary assumptions should be clearly listed. For example, do we need slater condition or strong duality for ensuring LAMBDA's training is successful?

### Typos
- Page 3: CMPDs --> CMDPs

**Summary Of The Paper:**

This paper proposed a method called LAMBDA for optimizing an agent's policy in constrained Markov decision processes. LAMBDA is a model-based approach leveraging Bayesian world models, which deals with reward optimistically and safety pessimistically. The authors demonstrate the effectiveness of SG6 tasks in Safety-Gym in terms of sample efficiency and satisfaction of safety constraint(s).

**Summary Of The Review:**

I think this paper is well-written and the key ideas would be useful for the research community. Though I have a minor concern regarding how much contribution exists for extending unconstrained MDP to constrained MDP settings, the overall contribution of this paper is sufficient; hence, I vote for acceptance.

---

> ### Author Response · Authors · 2021-11-16
> **Response to Reviewer HHZf**
>
> Thank you for your review on our paper! Please find below our response.
>
> >New and promising paradigm. Utilizing Bayesian world models for CMDPs is indeed powerful tool for RL community. Since humans would decide their behavior using such way of thinking in real life; thus, the idea for using Bayesian world models for CMDPs would be worthwhile to study
>
> Thank you for this interpretation of our paper!
>
> >If it is open-sourced, it would be quite helpful for the research community.
>
> We will de-anonymize the source-code (which is currently available [here](https://anonymous.4open.science/r/la-mbda/README.md)) and publish it on a github repository once possible.
>
> >Incorporating optimism for reward and pessimism for safety is not a new idea, which is not an essential contribution.
>
> We definitely agree with you that pessimism for safety is an intuitive (yet powerful) idea. However, the novelty in our proposed method is the way this idea is framed within Bayesian model-based reinforcement learning and constrained policy optimization.
> Furthermore, we show that this method scales to deep reinforcement learning applications.
>
>  >The world model (i.e., RSSM) is trained from scratch in the authors' experiments? My understanding is that, in the experiments presented in the paper, the world model is trained from scratch.
>
> Yes, it is trained from scratch (see Algorithm 2).
>
> >If a pre-trained model is obtained, the training of the agent will become much safer. Is my understanding correct?
>
> In preliminary experiments, we observed that pre-training the model improved the cost regret metric. We wish to further investigate this idea, towards safe exploration.
>
> Let us know if you have any further questions, we would be very happy to answer.

---

### Official Review · Reviewer_Z22c · 2021-10-31

**Correctness:** 4
**Technical Novelty And Significance:** 3
**Empirical Novelty And Significance:** 3
**Recommendation:** 8
**Confidence:** 3

**Main Review:**

## Strengths

- The paper is very well written and easy to follow.

- I believe that using model-based approaches to increase sample efficiency in safety-sensitive applications is very relevant. I also find interesting the use of a formulation for the transition model that is optimistic w.r.t. to the main objective while being pessimistic in terms of constraint satisfaction to be a very interesting idea.

- The proposed algorithm and design choices are clearly presented and motivated.

- The related work section is well organised and seemed sufficient to me.

- The experiments are in my opinion sufficient in volume and in supporting the claims made by the authors.

- The code for the experiments is publicly available online and many implementation details and hyperparameters are listed in the appendix which should allow for reproducibility of this work.

## Questions and weaknesses

1. The Augmented Lagrangian method presented in Section 4.1 is described in the Methods sections for the proposed algorithm (LAMDBA) rather than in the background section. Are the authors framing the use of Augmented Lagrangian for solving CMDPs (as opposed to the regular Lagrangian relaxation) as a contribution of this work? Was this version of Lagrangian method used in previous work on Constrained RL? If it has been used before, I believe citations to such previous work should be added. If this is presented as a contribution, I believe it would have been interesting to show comparisons between the augmented and regular lagrangian methods in the experiment section.

2. I believe that it would be more clear and impactful to show the comparison of sample efficiency between LAMBDA and the model-free algorithms in a table rather than simply in text (section 5.1). It is in my opinion one of the most important claims and results of the presented work and thus these results should be very easily retrievable when going through the manuscript.


**Summary Of The Paper:**

The authors present LAMBDA, a novel algorithm for learning in CMDPs. The approach seeks to improve on the sample efficiency of previous algorithms by learning a probabilistic model of the environment. This model is then used to train a policy to maximise the return and respect the constraints by backpropagating through imaginary trajectories. The uncertainty of the model can further be used to steer the agent towards for optimistic or pessimistic behavior. The method is benchmarked against one online planning method and previous model-free approaches on the SG6 set of tasks of the Safety Gym environment. The results show that the proposed method compares favourably to competing algorithms, particularly in terms of constraint satisfaction and sample efficiency.

**Summary Of The Review:**

I believe that this is a good paper making interesting and important contributions to the area of constrained RL and that it is ready to be shared with the community.

---

> ### Author Response · Authors · 2021-11-16
> **Response to Reviewer Z22c**
>
> Thank you for your review and accurate summary of our paper! Please find our responses to your questions below.
>
> >I believe that it would be more clear and impactful to show the comparison of sample efficiency between LAMBDA and the model-free algorithms in a table rather than simply in text (section 5.1).
>
> That's a great idea. We added a plot to the appendix that compares the amount steps it takes to the model-free methods to reach to LAMBDA's performance.
>
>
> > I believe that this is a good paper making interesting and important contributions to the area of constrained RL and that it is ready to be shared with the community.
>
> Thank you!
>
> >Are the authors framing the use of Augmented Lagrangian for solving CMDPs (as opposed to the regular Lagrangian relaxation) as a contribution of this work? Was this version of Lagrangian method used in previous work on Constrained RL? If it has been used before, I believe citations to such previous work should be added.
>
> We do not claim using the Augmented Lagrangian for constrained RL as a contribution but rather use it within our model-based constrained policy optimization framework. While the latter is our main contribution. We added a citation (Li et al. 2021) of a recent work comparing the Augmented Lagrangian with (regular) Lagrangian relaxation applied to CMDPs. Let us know if this addressed your concern.
>
> References:
> * Jingqi Li, David Fridovich-Keil, Somayeh Sojoudi,and Claire J. Tomlin. Augmented Lagrangian Method for Instantaneously Constrained Reinforcement Learning Problems. 2021.

---

> > ### Comment · Reviewer_Z22c · 2021-11-18
> > **Follow up**
> >
> > > Let us know if this addressed your concern.
> >
> > It does, thank you!

---

### Official Review · Reviewer_E48X · 2021-11-02

**Correctness:** 3
**Technical Novelty And Significance:** 2
**Empirical Novelty And Significance:** 3
**Recommendation:** 6
**Confidence:** 4

**Main Review:**

Review of: CONSTRAINED POLICY OPTIMIZATION VIA BAYESIAN WORLD MODELS

The paper presents an argument that model-based RL should have special value in situations when safety and efficiency is a concern. The paper raises interesting questions, but ultimately cannot compellingly make the case for the proposed approach. Specifically, the suitability of the model would seem to be a critical issue. However, the paper does not systematically address where such a model would come from, or how errors in the supplied model might affect performance. This is a critical issue in differentiating between model-free and model-based RL. Also, the argument in the paper lacks structure in a way that renders the contribution unclear. What is the relationship between the contributions outlined at the beginning of the paper and the questions outlined in the experiment section? Which aspects of the proposed approach are novel? How does the novelty of the proposed approach contribute to performance? Ultimately the lack of a clear source for the model that explains why performance may be improved is a critical limitation for the current work.

Minor comments:
- Figure one is unhelpful. The vertical axis is not labeled and three different scale metrics are plotted side by side. Also, what is SG6 and why should we care about it?
- It is unclear how one could experience unsafe events through model-generated trajectories, unless one could already solve the problem.
- The contributions do not make sense in the context of the introduction. Why are these problems to be solved?
- When you say "harness" what does that mean?
- "Curi et al. (2021) and Derman et al. (2019) also take
a Bayesian optimistic-pessimistic perspective to find robust policies. However, these approaches do not use CMDPs and generally do not explicitly address safety." Did you compare with them?
- "revolves around the repetition"
- The definition of model based reinforcement learning could be much more precise
- Please introduce acronyms prior to use
- "MBRL achieves superior sample efficiency compared to its model-free counterparts" This claim could be more precise.
- "We utilize"
- Please justify why we should consider the predictive distribution to be differentiable.
- "Thus, it is possible ..." repetitive.
- "in the model, to identify" --> "in the model to identify"
- The usefulness of uncertainty quantification depends on the degree to which the model is accurate with respect to the world.
- The logic of the experimental design and relation to the main questions was difficulty to understand given the high degree of overlap between the current method and previous work.

**Summary Of The Paper:**

The manuscript introduces LAMBDA a Bayesian model-based policy optimization algorithm that adheres to supplied safety constraints. The approach relies on the model to simulate trajectories and therefore improve efficiency of learning and effectiveness of safety. Experiments on SG6 compare the proposed algorithm to previous approaches and ablations of LAMDA. Based on experimental results, authors conclude that LAMDA is more efficient and effective for learning and safety.


**Summary Of The Review:**

Interesting paper focusing on model-based RL that does not address where the model comes from or the consequences of misspecification.

---

> ### Author Response · Authors · 2021-11-16
> **Response to Reviewer E48X**
>
> Thank you for reviewing our paper!
> Please find our detailed responses to the concerns raised. Should you have any further questions or if this did not address your concerns, please feel free to reach out again.
>
> >However, the paper does not systematically address where such a model would come from, or how errors in the supplied model might affect performance.
>
> We _learn_ a flexible (NN-based) model from scratch during exploration (see Algorithm 2). We demonstrate the utility in our experiments, where we extensively compare with model-free methods.
>
> >The usefulness of uncertainty quantification depends on the degree to which the model is accurate with respect to the world.
>
> Quantifying the _epistemic_ uncertainty is useful _throughout_ training. In particular, at the early stages of training, where models tend to be _less accurate_ (due to lack of data), quantifying this uncertainty becomes even more crucial. This is mainly for two reasons: **(1)** the uncertainty can be used to direct the agent to state-action pairs that have not been explored enough before (see, e.g., Curi et al. 2020, Auer et al. 2007); and **(2)** it 'signals' the agent when it reaches to state-action pairs with which the worst case scenario fails to satisfy the constraints. We agree that the _aleatoric_ uncertainty is useful when the model is accurate. We would be very happy to make this point clearer if there is a specific section of the paper that was misleading in this regard.
>
> >"Curi et al. (2021) and Derman et al. (2019) also take a Bayesian optimistic-pessimistic perspective to find robust policies. However, these approaches do not use CMDPs and generally do not explicitly address safety." Did you compare with them?
>
> Making such a comparison would not necessarily yield very valuable insights as the aforementioned methods use optimism-pessimism to solve a different set of problems (that come with a different set of metrics), i.e., they do not claim to address safety but rather adversarial robustness.
>
> >Figure one is unhelpful. The vertical axis is not labeled and three different scale metrics are plotted side by side.
>
> Figure 1 is meant to summarize our experimental results. To improve it, we rescaled the results such that the methods are scaled  _per metric_. We also added another section in the appendix that explains this better. Let us know if this helps.
>
> >What is the relationship between the contributions outlined at the beginning of the paper and the questions outlined in the experiment section?
>
> We revised these two paragraphs trying to associate better between them within the space limits. Please let us know if this resolves your concern.
>
> >It is unclear how one could experience unsafe events through model-generated trajectories, unless one could already solve the problem.
>
> We could not fully understand your meaning by saying "solve the problem". We were unsure if you meant solve CMDPs or solve the problem of predicting whether an event is safe or not.
> We want to emphasize that the agent does experience unsafe events also while interacting with the real environment. These events are then learned by the model for further use, even before the CMDP is solved.
> In particular, we are not claiming to solve the problem of _safe exploration_.
>
> >The contributions do not make sense in the context of the introduction. Why are these problems to be solved?
>
> The first two contributions translate into the main ingredients with which LAMBDA is able to solve CMDPs. In the first contribution we propose a novel approach for constrained policy optimization with 'through-model' gradients (in contrast to REINFORCE gradients (Williams et al. 1992)). This is already a crucial step for improving sample efficiency for constrained policy optimization. In the second contribution, by taking a Bayesian perspective, we allow the agent to balance between safety and exploration (this is especially evident in figure 11 in the PointButton1 task.) These two contributions are the keystones in our method for solving the challenges presented in the introduction (i.e., solving CMDPs, efficiently). Finally, the last contribution brings about a new state-of-the-art baseline that other model-based approaches can use while trying to solve CMDPs for safety.
>
> References:
> * Sebastian Curi, Felix Berkenkamp, and Andreas Krause. Efficient model-based reinforcement learning through optimistic policy search and planning, 2020.
> * Peter Auer and Ronald Ortner. Logarithmic online regret bounds for undiscounted reinforcement learning. In B. Schölkopf, J. Platt, and T. Hoffman (eds.), Advances in Neural Information Processing Systems, volume 19. MIT Press, 2007.
> * Williams, R.J. Simple statistical gradient-following algorithms for connectionist reinforcement learning. Mach Learn 8, 229–256 (1992).

---

### Official Review · Reviewer_11DZ · 2021-11-06

**Correctness:** 3
**Technical Novelty And Significance:** 3
**Empirical Novelty And Significance:** 3
**Recommendation:** 8
**Confidence:** 4

**Main Review:**

Overall, this paper this paper is building on a lot of different recent ideas. While I'm well versed in CMDPs and constrained optimization in general, I had to do some additional reading to understand the remaining pieces. This is not a bad thing, on its own and in fact your paper made me learn about a lot of new ideas. Thanks for that!

A couple of points: your decision to use SWAG, is interesting, especially from a computational perspective. Now from what I undertand from the original paper, this may a give you a crude approximation to the true posterior. What do we know about SWAG in the RL context? Does it still give you the right approximation? I presume that because your applying SWAG at the level of the dynamics themselves, you are probably dealing with a supervised learning problem, in which case prior results still hold. Is that the case? Because otherwise, I may see some potential complications when using SWAG in a setting where you do derivative estimation (reparameterization or score function) due to the additional estimation noise.

When I look over your experiment section, I would have like to seem some results on this component alone. I would want you to show me that your SWAG approximation does work. You must have gone through this exercise yourself when developing your approach and coding it up, so I'm sure you thought of a way to assess it.

Regarding the optimization method itself, I'm quite happy with your decision to go for an augmented lagrangian approach. However, I would have like you again to weight the pros and cons of this method, and convince me empirically that it's the right choice. A naive baseline here would be to do gradient descent-ascent/first order Lagrangian method/Arrow-Hurwicz; a fancier one would be SQP/interior point method. I was a bit confused too when I first read your algorithm description because it doesn't look like a typical augmented Lagrangian method due to the fact that you use a quadratic penalty for the magnitude of the lagrange multipliers themselves and not for the constraints themselves. I had to open up Nocedal and search for the exact algorithm that your are describing (p. 523 "unconstrained formulation" in my copy). There I found a description that looks a lot like what you have, but with better explanation as to how you obtain this (non-standard) variant on the augmented lagrangian. Also interesting note in Nocedal: "Unlike the bound-constrained and linearly constrained formulations, however, this unconstrained formulation is not the basis of any widely used software packages, so its practical properties have not been tested.", which explains my confusion.



**Summary Of The Paper:**

This paper is about solving CMDPs in an uncertainty-informed model-based fashion, without going through the usual LP route. The blueprints for this paper is the following:

1. Take a standard CMDP formulation
2. Express the objective and constraints in a UCRL (Auer, 2009) fashion (joint maximization over the set of possible dynamics),
3. Represent and learn the dynamics as a  Recurrent State Space Model from Hafner et al. (2019)
4. Estimate the objective and constraints themselves  (which involve a max operator) by sampling, and where the Bayesian posterior over the "world model" parameters is computed by SWAG (Maddox et al. 2019), where you average out the iterates generated over a trajectory of stochastic gradient descent
5. Using 3 and 4, compute a step of Augmented Lagrangian (a variant of) to solve the CMDP in 2.

The authors provide motivation as to why the UCRL aspect is important, and how the underlying optimism principle gives rise to persimism in the constraints, which is important from a "safety" perspective. This gives rise to an interesting tension (perhaps more "balance") between optimism in maximizing the objective while remaining cautions in abiding to the constraints.


**Summary Of The Review:**

I like the set of methods proposed by the authors, although motivation is lacking for why the particular sub-components have been chosen. This seems like a believable/feasible approach to Bayesian RL, and it inspires me to try things out along those lines. The criticism in my main review is about performance assessment of the SWAG component as well as the lack of motivation for the specific  (non-standard, but interesting!) optimization method.

---

> ### Author Response · Authors · 2021-11-16
> **Response to Reviewer 11DZ**
>
> Thank you for your review and and detailed comments! You can find below our response to some of your questions. Please let us know if you have any further concerns or suggestions.
>
> >While I'm well versed in CMDPs and constrained optimization in general, I had to do some additional reading to understand the remaining pieces. This is not a bad thing, on its own and in fact your paper made me learn about a lot of new ideas. Thanks for that!
>
> We are glad to hear that and hope that our findings could reach more people in this way.
>
> >I would want you to show me that your SWAG approximation does work.
>
> We agree that we did not show that SWAG provides good posterior approximation uniformly over the state-action space which might be needed in some applications.
> However, we only need SWAG to get useful uncertainty estimates for optimism and pessimism. In our experiments we indeed compare with an approach that averages across posterior samples ('greedy-exploitation') which can be interpreted as not using the provided uncertainty of SWAG. Note that our approach naturally allows using alternative approximate inference methods such as variational-inference, deep ensembles, etc; we chose SWAG for computational efficiency and simplicity. We agree however, that exploring different posterior approximation methods within RL is a topic that worth further research.
>
> >Regarding the optimization method itself, I'm quite happy with your decision to go for an augmented lagrangian approach. However, I would have like you again to weight the pros and cons of this method, and convince me empirically that it's the right choice. A naive baseline here would be to do gradient descent-ascent/first order Lagrangian method/Arrow-Hurwicz; a fancier one would be SQP/interior point method.
>
> Thank you for pointing this out. Our main motivation in choosing the Augmented Lagrangian was for two reasons: **(1)** its known good convergence properties (e.g., see, Rockafellar, 1974) and, **(2)** the fact that it is not inherently restricted to the computation of Hessians (as in SQP methods and some interior-point methods), which typically does not scale well to any but small neural-networks. Making a comparison of how the different methods perform in solving CMDPs deserves a dedicated experimental setup which we believe is out of scope for this paper. We added however a citation to a paper that could be a good starting point for such an experiment (Li et al. 2021). We hope that this addition helps motivating our decision better.
>
> >I was a bit confused too when I first read your algorithm description because it doesn't look like a typical augmented Lagrangian method...
>
> A derivation that views the problem from a different perspective but leads to the same expressions can be found in Bertsekas (1996) p. 158-162 (especially eqs. 10 and 17).
>
> References:
> * R. T. Rockafellar, Augmented lagrange multiplier functions and duality in nonconvex programming, SIAM J. Control Optim., 12 (1974), pp. 268--285.
> *Jingqi Li, David Fridovich-Keil, Somayeh Sojoudi,and Claire J. Tomlin. Augmented Lagrangian Method for Instantaneously Constrained Reinforcement Learning Problems. 2021.
> *Bertsekas, D. P. (1996). Constrained Optimization and Lagrange Multiplier Methods (Optimization and Neural Computation Series). Athena Scientific.

---

### Decision · Program_Chairs · 2022-01-20

**Decision:**

Accept (Spotlight)

**Comment:**

The paper describes a new model-based RL technique for constrained MDPs based on Bayesian world models.  It improves sample efficiency and safety. The reviewers are unanimous in their recommendation for acceptance.  This represents an important advance in RL.  Great work!